# AT4TS : Autotune for Time Series Foundation Models

**Shivani Tomar**  *shivani.tomar1@ibm.com*
*Trinity College Dublin*
*IBM Research, Ireland*

**Seshu Tirupathi**  *seshutir@ie.ibm.com*
*IBM Research, Ireland*

**Radu Marinescu**  *radu.marinescu@ie.ibm.com*
*IBM Research, Ireland*

**Elizabeth M. Daly**  *elizabeth.daly@ie.ibm.com*
*IBM Research, Ireland*

**Ivana Dusparic**  *duspari@tcd.ie*
*Trinity College Dublin*

**Reviewed on OpenReview:** *https://openreview.net/forum?id=U54YyLn8MX*

## Abstract

Foundation models have been successfully adapted to the task of time series forecasting due to their ability to capture long-range dependencies, as demonstrated in the field of Natural Language Processing (NLP). However, effectiveness of applying these pre-trained time series foundation models (TSFMs) in the target domain is limited due to the need for hyperparameter optimization to match the characteristics of the target domain. To address this limitation, we propose a novel algorithm AT4TS: Autotune for Time Series Foundation Models that aims to efficiently automate the process of selective fine-tuning of pre-trained TSFMs for a given target domain. Our approach helps remove the tedious task of accurately configuring the tunable hyperparameters required to selectively update parameters to enhance predictive performance on unseen out-of-domain target datasets. AT4TS has been validated through diverse pre-trained models like Chronos and Tiny Time Mixers (TTM), fine-tuning strategies like Low Rank Adaptation (LoRA) and custom fine-tuning and state-of-the-art hyperparameter optimization (HPO) methods. Extensive experimental results on real-world benchmark datasets demonstrate that AT4TS efficiently identifies the optimal configuration of tunable hyperparameters for autotuning TSFMs. We show improvements as high as 20.55% and 45.34% for one of the out-of-domain datasets compared to zero-shot pre-trained models for Chronos and TTM respectively.

## 1 Introduction

Time series forecasting has always been critical for decision-making across various domains including retail, smart grids, healthcare, finance, weather, traffic control among others (Peterson, 2017) (Hernandez et al., 2014). The remarkable success of LLMs in broad domains like Computer Vision (CV) and Natural Language Processing (NLP) (Vaswani et al., 2017) (Wen et al., 2022) has prompted researchers to adapt these models to the task of time series forecasting (Wu et al., 2022; Garza & Mergenthaler-Canseco, 2023; Gruver et al., 2024) given their ability to capture long range dependencies present in time series data. These models are pre-trained on vast amounts of data spanning a multitude of domains leveraging the general purpose representations learnt in the process. However, to excel on the domain specific downstream task it becomes important to fine-tune these models on datasets from the target domain (Wen et al., 2022). Such adaptation

is generally achieved via fine-tuning, which involves updating all the parameters of the pre-trained model (Jin et al., 2023; Bommasani et al., 2021; Lv et al., 2023). Given the large number of parameters that are originally trained, full fine-tuning becomes an operational challenge if we wish to adapt these models to the target domain.

Recent studies have shown that the computational burden and inefficiencies associated with full fine-tuning can be mitigated by adapting only a small percentage of the parameters in addition to the pre-trained model for each task which greatly enhances the operational efficiency of these models (Hu et al., 2021). This helps in learning the specific characteristics of the target-domain while also retaining the broad understanding of general time series data acquired during the pre-training phase. These techniques are termed as Parameter-Efficient Fine-Tuning (PEFT). One of the popular PEFT techniques, LoRA (Low Rank Adaptation)(Hu et al., 2021) trains only selective dense layers in a neural network while keeping the pre-trained weights frozen. This approach has been widely applied to various domains such as medical imaging, video text generation and speech synthesis (Balne et al., 2024) to name a few. However, its application to time series is under explored.

AutoML, which involves automating the process of composing and parameterizing ML algorithms to maximize a specific metric such as model accuracy on a given dataset has been widely used to improve the accuracy of traditional machine learning and deep learning models (He et al., 2021). Leveraging this knowledge, auto-tuning of pre-trained foundation models for a given target domain can potentially lead to improvement in the accuracy of these models compared to traditional fine-tuning with fixed hyperparameters of the fine-tuning algorithms.

To this end, we propose a novel algorithm called AT4TS : Autotune for Time Series Foundation Models, which achieves efficient fine-tuning of pre-trained TSFMs through the integration of parameter-efficient fine-tuning techniques along with hyperparameter optimization of the fine-tuners to improve the performance of time series foundation models in the target domain. We illustrate our algorithm using two different categories of TSFMs: Chronos, a Transformer-based architecture and TTM, a lightweight non Transformer-based model. In particular, to achieve an efficient implementation of autotuning, we adopt the classical Limited Discrepancy Search (LDS) algorithm introduced by (Harvey & Ginsberg, 1995) to optimize the hyperparameter selection process. This algorithm is essentially a depth-first search strategy that identifies new set of solutions by iteratively increasing the number of discrepancy values where the discrepancy refers to the number of variables in the current configuration that differ from their values in the initial configuration. We further establish the robustness of AT4TS by using it with other hyperparameter optimization methods, namely, Hyperopt (Bergstra et al., 2015) and Bayesian Optimization and Hyperband (BOHB) (Falkner et al., 2018). The novel contributions of this paper include:

- Automated fine-tuning exploring tunable hyperparameter configurations for fine-tuners of pre-trained time series foundation models to find the optimal hyperparameters for the target domain. To the best of our knowledge, this is the first paper to explore the potential of autotuning TSFMs.

- Generalized applicability of AT4TS algorithm across disparate pre-trained models (transformer based and non-transformer based), fine-tuning strategies (LoRA and custom fine-tuning) and HPO methods (LDS, Hyperopt and BOHB).

- Extensive tests across a suite of out-of-domain real-world benchmark datasets to compare the performance of autotuned TSFMs, traditional fine-tuning strategies, and zero-shot pre-trained models.

## 2 Related Work

With the recent success of foundation models in the domains of NLP and CV, there has been growing research for leveraging their potential for time series analysis particularly for long-term forecasting. To this end, TSFMs aim to accomplish the zero-shot generalization capabilities across a diverse spectrum of downstream tasks, thereby, minimizing the need for explicit task-specific model development. Furthermore, TSFMs hold the promise of attaining superior performance through domain specific fine-tuning catering to a wide range of applications. Authors in (Liang et al., 2024), propose a comprehensive taxonomy offering a thorough

understanding of the rapidly evolving landscape of TSFMs. Based on their methodological categorization, most of the popular TSFMs are: Transformer-based, non Transformer-based and diffusion-based models. We briefly discuss the first two categories of TSFMs in this section as we illustrate our approach using one model from each category.

**Transformer-based TSFMs:** In the first category, authors in (Zhou et al., 2021) (Wu et al., 2021) (Zhou et al., 2022) employ transformer-based architectures to capture long-term dependencies in time series data. They further handle the challenges of quadratic time complexity and high memory usage through self-attention and auto-correlation mechanisms. PatchTST (Nie et al., 2022) on the other hand, introduced two key components : segmentation of time series into patches serving as input tokens to the transformer along with channel-independence overcoming the challenge of high memory usage of attention maps. Yet another pioneering work in this area is Lag-Llama (Rasul et al., 2023) inspired by the LLaMA (Touvron et al., 2023) LLM which utilizes a simple decoder-only transformer architecture for time series forecasting by using lagged features as covariates. Another line of work, LLMTime (Gruver et al., 2024), uses time series as strings with careful pre-processing specific to the given LLMs' tokenizer. Likewise, Chronos (Ansari et al., 2024) employs the encoder-decoder transformer architecture from the T5 (Raffel et al., 2020) LLM family requiring minimal modifications i.e, tokenization through scaling and quantization.

**Non Transformer-based TSFMs:** Non-Transformer based models use diverse architectures to efficiently process temporal patterns. One groundbreaking contribution in this category is a relatively smaller model (1-5 million parameters) called Tiny Time Mixers (TTM) (Ekambaram et al., 2024). Build upon TSMixer architecture, it uses light-weight MLPMixer blocks interleaved with simple gated attention outperforming many foundation models with significantly larger model parameter sizes.

**Parameter Efficient Fine Tuning (PEFT):** In order to make TSFMs more accessible and adaptable to facilitate their wider application to real-world downstream tasks, they can be directly deployed to leverage the general temporal knowledge acquired in the pre-training phase. However, in operational deployments, domain specific target data is used to fine-tune these models to further enhance their zero-shot performance. Most commonly, PEFT techniques have been proposed in NLP and CV for fine-tuning a subset of parameters in various downstream tasks. One of them is LoRA (Hu et al., 2021) which adds trainable low-rank matrices into transformer layers to approximate the weight updates while keeping the existing weights frozen. The application of LoRA in the domain of fine-tuning TSFMs remains largely unexplored due to the rapidly evolving nature of this field and hence, motivates this work. The authors in (Gupta et al., 2024) investigate the impact of LoRA based fine-tuning across popular TSFMs which relates to our work, however, they do not attempt autotuning TSFMs using different parameter efficient fine-tuning techniques which makes our contribution novel.

**Hyperparameter Optimization (HPO):** Hyperparameter optimization (HPO) can potentially act as an important component in searching for optimal hyperparameters for selectively fine-tuning the pre-trained models. Bayesian optimization (BO) has been the state-of-the-art method for optimizing hyperparameters for neural networks (Eggensperger et al., 2013) (Feurer et al., 2015). It relies on probabilistic modeling of the objective function given the observed data points. Other methods include Hyperband (Li et al., 2022) which is a bandit based method that adaptively allocates different budgets $b$ and utilizing successive halving to find the best out of a set of randomly sampled configurations. BOHB leverages the benefits of both BO and HB to achieve faster convergence to the most optimal configurations. Hyperopt is based on bayesian optimization that uses Tree-of-Parzen-Estimators (TPE) (Bergstra et al., 2011) algorithm for carrying out hyperparameter optimization for ML algorithms.

**Autotuning methods in other domains:** Autotuning has not been explored in the context of time series forecasting using TSFMs. However, the importance of obtaining the best hyperparameters for fine-tuning has been studied extensively across other domains. The authors in (Liu & Wang, 2021) discuss the impact of automated HPO methods on finetuning pre-trained language models. They systematically compare automated HPO methods with grid search and analyse the failure scenarios in HPO for fine-tuning. Another work (Tribes et al., 2023) explores HPO using blackbox optimization algorithms such as NOMAD and

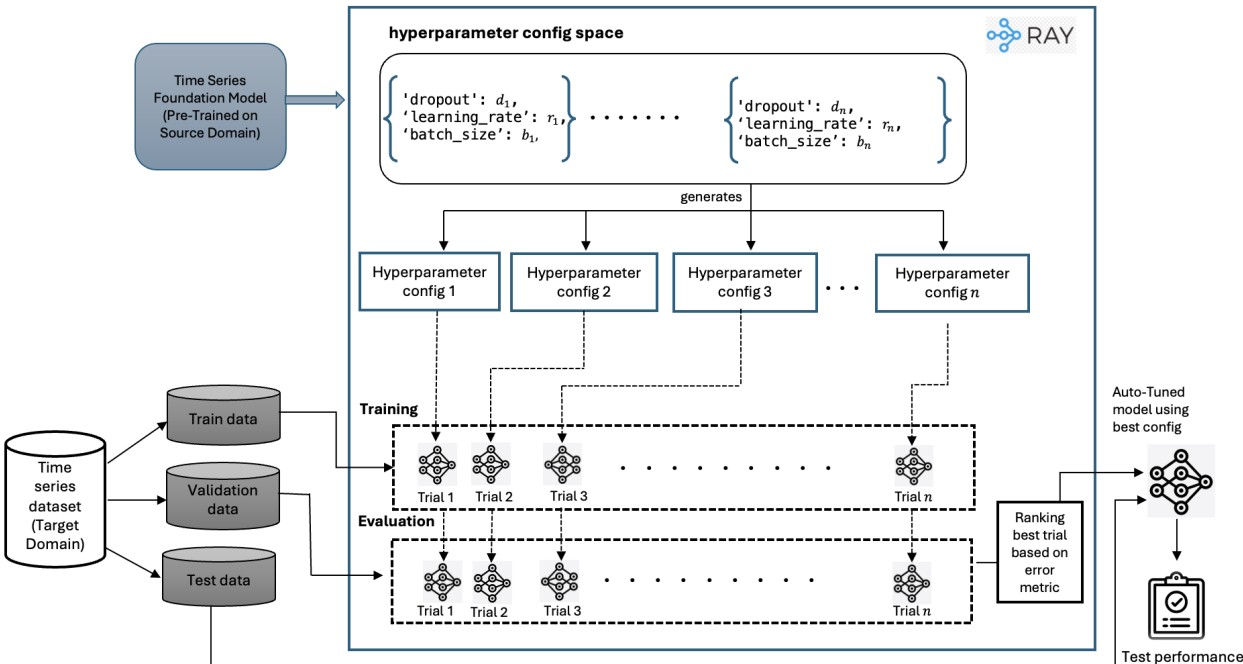

Figure 1: Architecture diagram showing the workflow of our algorithm, AT4TS using Hyperparameter Optimization and Ray Tune for parallelization of the tune trials.

demonstrate improved performance of fine-tuned LLMs on downstream tasks. (Li et al., 2020) suggest best practices to set hyperparameters for fine-tuning of models in image domain tasks. They further demonstrate that key hyperparameters such as effective learning rate, momentum and weight decay are dependent on the similarity between the source and target domains.

## 3  Methodology

In this section, we first define the problem statement and then present a detailed description of the AT4TS design approach we adopted using selective fine-tuning: LoRA and custom fine-tuning coupled with LDS as the search algorithm.

**Problem Statement**    Given a multivariate time series $X \in \mathbb{R}^{c \times n}$ of length $n$, number of channels/variables $c$ and context length $cl$ where $1 < cl < n$, the forecasting task is to predict future values $Y \in \mathbb{R}^{c' \times h}$ where $c'$ denotes the number of channels being forecasted and $h$ is the forecast horizon.

To solve the forecasting task described above while also demonstrating the contribution of this work through their application, we employ two TSFMs: Chronos (Ansari et al., 2024), a transformer-based model which handles univariate datasets and TTM (Ekambaram et al., 2024), a non-transformer-based model built for mutivariate data setting. These models have been pre-trained on a large collection of publicly available time series datasets from varied domains. However, since these models are trained on a broad spectrum of time series data, their performance on a specific task which in our case is the unseen target dataset may not always be optimal.

**Chronos with PEFT:**   We implement AT4TS using Chronos with PEFT. LoRA (Hu et al., 2021), a prevalent PEFT technique helps in integrating domain specific knowledge to the pre-trained model by fine-tuning a minimal number of weights. This is achieved in a storage and compute-efficient manner by constraining the

---

**Algorithm 1** AT4TS algorithm using selective fine-tuning and LDS

---

    **Input:** Pre-trained Model $M$, target dataset $X$ split into $X_{\text{train}}$, $X_{\text{val}}$ and $X_{\text{test}}$, search space $\mathbf{Y} = \{Y_1, ..., Y_n\}$, SEARCH with maximum discrepancy $\delta$

    **Output:** Optimally tuned hyperparameters $Y_{opt}$ and Autotuned Model $M_{Y_{opt}}$

 1: Define hyperparameter search space $\mathbf{Y} = \{Y_1, ..., Y_n\}$

 2: Execute SEARCH : Initialize $\mathbf{y^0}$ to default hyperparameters and let $\mathbf{y^*} \leftarrow \mathbf{y^0}$

 3: **for** all $\theta = 1, \ldots, \delta$ **do**

 4:     SEARCH($\mathbf{y^*}$, $\mathbf{Y}$, $\theta$, 1)

 5: **end for**

 6: **return** $\mathbf{y^*}$, $M^*$

 7: **procedure** SEARCH($\mathbf{y}$, $\mathbf{Y}$, $\theta$, $i$)

 8:     **if** $\theta == 0$ **or** $i > |\mathbf{Y}|$ **then**

 9:         $Y_{opt}, M_{Y_{opt}} \leftarrow$
            SCORE($\mathbf{y}$, $X_{\text{train}}$, $X_{\text{val}}$, $M$)

10:         **return** $Y_{opt}, M_{Y_{opt}}$

11:     **else**

12:         **for** all values $y \in D(\mathbf{Y}[i])$ **do**

13:             **if** $\mathbf{y}[i] == y$ **then**

14:                 $z \leftarrow$ SEARCH($\mathbf{y}, \mathbf{Y}, i+1, \theta$)

15:             **else**

16:                 $\mathbf{y'} \leftarrow \mathbf{y}$;   $\mathbf{y'}[i] \leftarrow y$

17:                 $z \leftarrow$ SEARCH($\mathbf{y'}, \mathbf{Y}, i+1, \theta-1$)

18:             **end if**

19:             **return** $z$

20:         **end for**

21:     **end if**

22: **end procedure**

23: **procedure** SCORE($\mathbf{y}$, $X_{\text{train}}$, $X_{\text{val}}$, $M$)

24:     $M \leftarrow$ TrainModel($\mathbf{y^*}$, $X_{\text{train}}$)

25:     score $\leftarrow$ EvaluateModel($M$, $X_{\text{val}}$)

26:     **if** score $>$ best_score **then**

27:         $M_{Y_{opt}} \leftarrow M$

28:         $Y_{opt} \leftarrow \mathbf{y^*}$

29:         best_score $\leftarrow$ score

30:     **end if**

31:     **return** $Y_{opt}, M_{Y_{opt}}$

32: **end procedure**

---

update ($\Delta W$) to the pre-trained weight matrix $W_0 \in \mathbb{R}^{d \times k}$ by representing it with a low-rank decomposition,

$$W_0 + \Delta W = W_0 + BA$$

where $B \in \mathbb{R}^{d \times r}$, $A \in \mathbb{R}^{r \times k}$, and the rank $r \leq \min(d, k)$. During fine-tuning, $W_0$ is frozen, while the weights of $A$ and $B$ are updated. We adapt the weight matrices corresponding to the self-attention module and the feed-forward layer modules of the transformer architecture.

**TTM with custom fine-tuning:** In the case of TTM, we adopt the fine-tuning technique as described in (Ekambaram et al., 2024) wherein the backbone of the model is frozen while only updating the weights of the TTM forecast head. This is because of the following three reasons: (1) this is the fine-tuning strategy used by TTM developers and therefore, makes it a credible comparison; (2) the size of these models is much smaller and so LoRA is not as efficient as shown by preliminary experiments we conducted; and (3) we want to demonstrate that AT4TS can be applied across different fine-tuning strategies.

**AT4TS using parameter efficient fine-tuning and LDS:** We design a novel algorithm to perform automated fine-tuning of time series foundation models described above using LoRA and custom fine-tuning methods in conjunction with LDS. This fine-tuning is achieved by using a distributed Ray-based framework[1]. Figure 1 shows the architecture diagram of our approach wherein we take a pre-trained model and specify a minimal subset of weights to be trained to adapt the model on the target dataset. The *hyperparameter config space* block corresponds to the hyperparameter search space which is defined as the domain of values that are explored and evaluated during the process of hyperparameter tuning. For instance, LoRA hyperparameters include learning rate, batch size, rank, scaling factor etc as shown in Table 2. The *hyperparameter config blocks 1 to n* correspond to trials which are executed concurrently in a distributed cluster. Each trial represents a model fine-tuned and evaluated on the train and validation split of the target dataset respectively. The most promising hyperparameter configuration is obtained by ranking the trials based on the chosen error or performance metric on the validation split. In the end, the fine-tuned model corresponding to the best-found hyperparameter configuration (also called the autotuned model) is evaluated on the test split of the target dataset.

Algorithm 1 outlines the steps involved in AT4TS. The algorithm starts with initialising the hyperparameter search space. We use Limited Discrepancy Search or LDS (Harvey & Ginsberg, 1995) to traverse this space effectively starting from an initial configuration of default hyperparameters. It should be noted that AT4TS provides the flexibility of using any other search algorithm and is not limited by LDS. Specifically, LDS takes as input a vector of variables $\mathbf{Y} = \{Y_1, ..., Y_n\}$ corresponding to the hyperparameters together with their domains of values $\mathbf{D} = \{D(Y_1), ..., D(Y_n)\}$ representing the hyperparameter search space to be explored and the maximum discrepancy value $\delta$ which limits the number of allowed variable-value assignment changes from the initial solution $\mathbf{y}^0 = (y_1^0, ..., y_n^0)$ where $y_i^0 \in D(Y_i)$ is a value in variable's $Y_i$ domain and outputs the next solution $\mathbf{y}^*$ based on the discrepancy value. Notice that LDS requires the variables to have finite domains of values, and therefore any continuous hyperparameter needs to be discretized. In addition, LDS is required to look for a reasonably good solution which is typically given by the default hyperparameter configuration here, hence, we set $\mathbf{y}^0$ to the default LoRA configuration for Chronos and default fine-tuning parameters in case of TTM. We begin with a discrepancy value $\theta$ of 1 and conduct an iterative search that allows to change the values of at most $\theta$ hyperparameters in the initial solution $\mathbf{y}^0$. We then increment $\theta$ until $\theta$ exceeds the maximum discrepancy value. Function SEARCH defined in line 7 performs the actual exploration of the hyperparameter search space limited by discrepancy $\theta$. This assists in incrementally searching around the default configuration compared to random exploration. For each configuration $\mathbf{y}'$ returned by LDS, we fine-tune the model in line 24 and evaluate it on the validation split to compute the error metric as outlined in the SCORE function in the algorithm. At the end, we get the best configuration $\mathbf{y}^*$ corresponding to the lowest error and the autotuned model $M^*$ in line 6 which is then evaluated on the held out test split.

For illustration, Figure 2 shows the search space explored by LDS with discrepancy value 1 (denoted by LDS(1)) for 3 dummy variables [A, B, C] with domain values $\{a_1, a_2\}$, $\{b_1, b_2\}$ and $\{c_1, c_2\}$, respectively. In this case, LDS(1) starts from the initial assignment of $\{a_1, b_1, c_1\}$ which corresponds to the leftmost blue leaf node and traverses the search space in a depth-first manner visiting only the blue leaf nodes.

The algorithm described above is then implemented in a distributed manner using Ray Tune (Liaw et al., 2018) which provides an open source framework for distributed model training and selection. Each configuration returned by LDS corresponds to the trials executed concurrently in a cluster. We use the default Ray tune scheduler which is first-in-first-out (FIFO) passing through the trial configurations without performing any early stopping.

## 4 Experimental Setup

In this section, we present the datasets used in the fine-tuning experiments along with the implementation details of the proposed AT4TS framework.

**Datasets** For our experiments, we use 10 univariate datasets from the Monash Time Series Forecasting Repository (Godahewa et al., 2021) for autotuning Chronos models. These datasets are part of the Bench-

---

[1]https://www.ray.io/

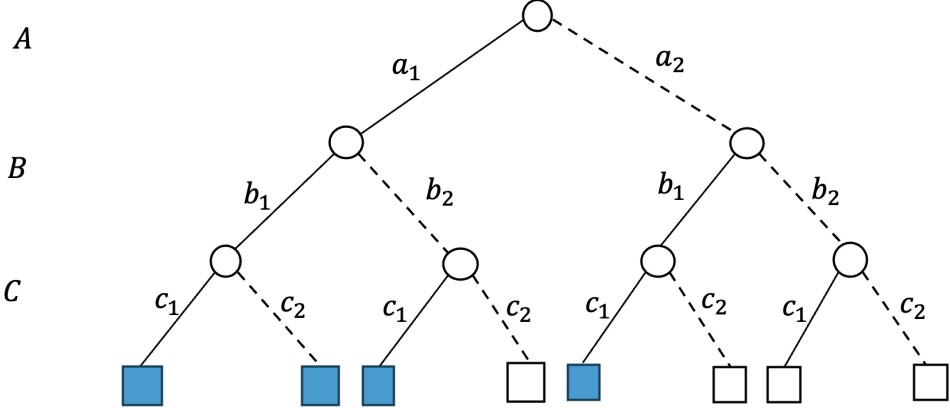

Figure 2: Example of the search space traversed by LDS with maximum discrepancy value = 1.

Table 1: Univariate Datasets used for the Chronos experiments.

| Dataset | Domain | Freq. | Num. Series | Series Length | Prediction Length (H) |
|---|---|---|---|---|---|
| Traffic | Transport | 1H | 862 | 17544 | 24 |
| Weather | Nature | 1D | 3010 | 14296* | 30 |
| ETT (Hourly) | Energy | 1H | 14 | 17420 | 24 |
| ERCOT Load | Energy | 1H | 8 | 154854 | 24 |
| Australian Electricity | Energy | 30min | 5 | 231052* | 48 |
| Exchange Rate | Finance | 1B | 8 | 7588 | 30 |
| FRED-MD | Economics | 1M | 107 | 728 | 12 |
| NN5 (Daily) | Finance | 1D | 111 | 791 | 56 |
| M5 | Retail | 1D | 30490 | 1562* | 28 |
| ETT (15 min.) | Energy | 15min | 14 | 69680 | 24 |

mark II datasets in (Ansari et al., 2024) used for zero-shot evaluation. Table 1 provides the details of the datasets used for the experiments. Each dataset is a collection of series where each series is split into train, validation and test instances. The * in the series length represents average length of the time series for 3 datasets, Weather, Australian Electricity and M5 which consists of variable length time series. The number of instances in validation and test label depend on the prediction horizon which is specific for each dataset. We use the GluonTS library to split the datasets using the same strategy as used in (Ansari et al., 2024). For each dataset, the train split includes all the data points from the beginning of the time series up until the last two prediction horizon lengths, which are held out for validation and testing respectively. We use these datasets as they have not been used in the pre-training phase of the Chronos T5 models. Similarly, Table 3 lists the datasets used for TTM in our experiments. All the datasets except MORE were used in (Ekambaram et al., 2024) for zero-shot/few-shot evaluation with no overlap with the pre-training datasets. These are multivariate benchmark datasets which do not contain any exogenous variables. To further validate the efficacy of our approach in an unseen target domain, we use a real-world wind energy dataset called MORE[2] data containing exogenous channels. This dataset uses wind park data for 18 months from 11 wind turbines (WT2 to WT11) in a wind park. The dataset consists of SCADA data from the sensors on the wind turbines combined with weather data. In our experiments, we only utilize a sample of this dataset with 9500 points. Since TTM can capture cross-channel correlations and exogenous variables, MORE data is an ideal fit for the autotuning application scenario targeted by our approach.

---

[2]https://github.com/MORE-EU/OpenData

Table 2: Chronos: LoRa Hyperparamater Search Space.

| Parameter Name | Range of values |
|---|---|
| alpha | {4, 8, 16, 32, 64} |
| dropout | {0.0, 0.05, 0.1} |
| rank | {2, 4, 8, 16, 32} |
| bias | {"none", "all", "lora_only"} |
| learning_rate | {0.0001, 0.001, 0.01} |
| batch_size | {4, 8, 16} |
| grad_accumulation_steps | {1, 4, 8} |

Table 3: Multivariate Datasets used for the TTM experiments.

| Dataset | Freq. | Series Length | Total Channels | Target variables | Exogenous variables |
|---|---|---|---|---|---|
| MORE | 1H | 9500 | 13 | 1 | 12 |
| ETTh1 | 1H | 17420 | 7 | 7 | N/A |
| ETTh2 | 1H | 17420 | | | |
| ETTm1 | 15min | 69680 | | | |
| ETTm2 | 15min | 69680 | | | |
| Weather | 10min | 52696 | 21 | 21 | |
| Electricity | 1H | 26304 | 321 | 321 | |

**Models** We used Chronos T5 models as the Transformer-based architecture for demonstrating our approach, AT4TS. These models, trained from scratch on time series data, are widely adopted open-source transformers that have achieved state-of-the-art (SOTA) performance in time series analysis. Chronos T5 models have been pre-trained and released in 5 sizes ranging from Tiny (16M), Mini (20M), Small (46M), Base (200M) and Large (710M) with number of model parameters in brackets. In our experiments, we focus on univariate time series forecasting as Chronos models are pre-trained for the univariate setting. We use the lightweight version of the models i.e. Mini in order to utilize minimal computational resources for demonstrating the applicability of our approach. We also used TTM which is a significantly smaller (1-5M) pre-trained model for effective zero/few-shot multivariate forecasting. TTM is composed of TSMixer architecture, based on MLP blocks and has 4 key components namely, TTM backbone, TTM decoder, Forecast head and the optional exogenous mixer known for capturing cross channel relationships and exogenous variables. They have been released as 3 variants: TTM-Base (1M), TTM-Enhanced (4M) and TTM-Advanced (5M) trained with context length, 512, 1024 and 1536 respectively. We specifically used the lightweight TTM-Base in our experiments. Our experiments validate that AT4TS is model agnostic and can be flexibly adapted to use both transformer and non-transformer based TSFMs.

Table 4: TTM: Custom Fine-Tuning Hyperparamater Search Space.

| Parameter Name | Range of values |
|---|---|
| head_dropout | {0.1, 0.4, 0.7} |
| batch_size | {8, 16, 32, 64} |
| learning_rate | {0.001, 0.0001} |

Table 5: Mean MASE scores obtained using Chronos T5 mini model in Zero Shot, Default LoRA Fine-Tuning and our approach, AT4TS with LDS.

| Dataset | Zero Shot | Default LoRA Fine Tuning | AT4TS(LDS) |
|---|---|---|---|
| Traffic | 0.853 ($\pm$0.0012) | 0.776 ($\pm$0.0015) | **0.747** ($\pm$0.0157) |
| Weather | 0.859 ($\pm$0.0032) | 0.848 ($\pm$0.0039) | **0.822** ($\pm$0.0043) |
| ETT (Hourly) | **0.795** ($\pm$0.0111) | 0.830 ($\pm$0.0179) | 0.797 ($\pm$0.0235) |
| ERCOT Load | 0.582 ($\pm$0.0107) | **0.566** ($\pm$0.0341) | **0.566** ($\pm$0.0341) |
| Australian Electricity | 0.965 ($\pm$0.0406) | 1.151 ($\pm$0.0300) | **0.832** ($\pm$0.0827) |
| Exchange Rate | 2.054 ($\pm$0.1561) | 1.871 ($\pm$0.0527) | **1.632** ($\pm$0.1756) |
| FRED-MD | **0.473** ($\pm$0.0105) | 0.508 ($\pm$0.0100) | 0.511 ($\pm$0.0092) |
| NN5 (Daily) | 0.648 ($\pm$0.0059) | **0.605** ($\pm$0.0012) | 0.620 ($\pm$0.0138) |
| M5 | 0.942 ($\pm$0.0004) | 0.926 ($\pm$0.0003) | **0.925** ($\pm$0.0005) |
| ETT (15 min.) | **0.709** ($\pm$0.0269) | 0.712 ($\pm$0.0172) | 0.727 ($\pm$0.0378) |

Table 6: Mean MASE scores obtained with Chronos T5 mini model using different HPO techniques in AT4TS and average rank (lower is better).

| Dataset | AT4TS | | |
|---|---|---|---|
| | BOHB | HyperOpt | LDS |
| Traffic | 0.768 ($\pm$0.0041) | 0.779 ($\pm$0.0495) | **0.747** ($\pm$0.0157) |
| Weather | **0.818** ($\pm$0.0037) | 0.824 ($\pm$0.0028) | 0.822 ($\pm$0.0043) |
| ETT (Hourly) | 0.798 ($\pm$0.0231) | **0.794** ($\pm$0.0036) | 0.797 ($\pm$0.0235) |
| ERCOT Load | 0.617 ($\pm$0.0301) | 0.602 ($\pm$0.0456) | **0.566** ($\pm$0.0341) |
| Australian Electricity | **0.738** ($\pm$0.0699) | 0.829 ($\pm$0.0257) | 0.832 ($\pm$0.0827) |
| Exchange Rate | 2.262 ($\pm$0.0568) | 2.143 ($\pm$0.5007) | **1.632** ($\pm$0.1756) |
| FRED-MD | 0.540 ($\pm$0.0560) | 0.554 ($\pm$0.0288) | **0.511** ($\pm$0.0076) |
| NN5 (Daily) | 0.601 ($\pm$0.0116) | **0.597** ($\pm$0.0090) | 0.620 ($\pm$0.0138) |
| M5 | 0.923 ($\pm$0.0005) | **0.922** ($\pm$0.0011) | 0.925 ($\pm$0.0005) |
| ETT (15 min.) | 0.694 ($\pm$0.0295) | **0.694** ($\pm$0.0121) | 0.727 ($\pm$0.0378) |
| Avg. Rank | 2.0 | **1.9** | **1.9** |

**Implementation Details** AT4TS has been implemented using Ray Tune (Liaw et al., 2018) and Transformers[3] libraries. It supports both Transformer and non-Transformer TSFMs as well as parameter efficient fine-tuning (backed by the PEFT[4] Library). The number and range of tunable hyperparameters are set based on the fine-tuning strategy being used. Tables 2 and 4 show the search space for LoRA hyperparameters and custom fine-tuning hyperparameters used in autotuning Chronos and TTM models respectively. We execute multiple trials selected using LDS for each dataset to find the best hyperparameter configuration and output the autotuned model corresponding to it. We limit the number of trials to 10 in case of Chronos and 15 for TTM to demonstrate the robustness of our approach in a resource-constrained environment. We used the maximum discrepancy value based on the number of hyperparameters to be tuned. In our case, we set this to 8 for Chronos and 2 for TTM. A lower value of maximum discrepancy involves a more focused search while higher values allow more broader exploration of the potential hyperparameter search space. To ensure a comprehensive evaluation across different fine-tuning settings in the case of Chronos, we use mean absolute scaled error (MASE) as the evaluation metric. Since the model produces probabilistic forecasts, the forecasted value for each datapoint is calculated as the median (0.5-quantile) of 20 samples, which is then used to calculate the metrics similar to (Ansari et al., 2024). The MASE scores are averaged across 5 runs. For TTM, we use mean squared error (MSE) as the standard error metric and report average MSE scores across five forecast lengths, $FLs \in \{24, 48, 60, 96, 192\}$ similar to (Ekambaram et al., 2024). The

---

[3]https://huggingface.co/docs/transformers
[4]https://huggingface.co/docs/peft

hyperparameters for default fine-tuning of TTM for ETTh/ETTm/weather/electricity are configured to the same values as used in (Ekambaram et al., 2024). For MORE dataset, we set the default values as follows: *learning rate*=0.001, *head dropout*=0.4 and *batch size*=64. The experiments were performed on a multi-node cluster environment using a combination of CPUs and A100 GPUs. In order to compare the performance of LDS search algorithm with other state-of-the-art HPO methods, we run AT4TS by plugging in Hyperopt and BOHB as the search algorithm. The detailed analysis of the results obtained are discussed in the next section.

Table 7: Mean MSE scores obtained using TTM-Base model in Zero Shot, Default Fine-Tuning and our approach, AT4TS.

| Dataset | Zero Shot | Default Fine Tuning | AT4TS |
|---|---|---|---|
| MORE | 0.247 ($\pm$0.1129) | 0.213 ($\pm$0.1044) | **0.135** ($\pm$0.0122) |
| ETTh1 | 0.348 ($\pm$0.0281) | 0.348 ($\pm$0.0303) | **0.343** ($\pm$0.0329) |
| ETTh2 | 0.245 ($\pm$0.0614) | 0.246 ($\pm$0.0639) | **0.241** ($\pm$0.0635) |
| ETTm1 | 0.286 ($\pm$0.0534) | 0.279 ($\pm$0.0513) | **0.259** ($\pm$0.0445) |
| ETTm2 | 0.152 ($\pm$0.0471) | 0.150 ($\pm$0.0464) | **0.144** ($\pm$0.0436) |
| Weather | 0.133 ($\pm$0.0353) | 0.133 ($\pm$0.0359) | **0.130** ($\pm$0.0347) |
| Electricity | 0.145 ($\pm$0.0305) | 0.137 ($\pm$0.0292) | **0.120** ($\pm$0.0179) |

Table 8: Mean MSE scores obtained with TTM using different HPO techniques in AT4TS and average rank (lower is better).

| Dataset | AT4TS | | |
|---|---|---|---|
| | BOHB | HyperOpt | LDS |
| MORE | 0.136 ($\pm$0.0129) | 0.137 ($\pm$0.0144) | **0.135** ($\pm$0.0122) |
| ETTh1 | **0.343** ($\pm$0.0329) | **0.343** ($\pm$0.0334) | **0.343** ($\pm$0.0329) |
| ETTh2 | 0.246 ($\pm$0.0681) | 0.244 ($\pm$0.0663) | **0.241** ($\pm$0.0635) |
| ETTm1 | 0.260 ($\pm$0.0449) | **0.259** ($\pm$0.0445) | **0.259** ($\pm$0.0445) |
| ETTm2 | **0.144** ($\pm$0.0434) | **0.144** ($\pm$0.0436) | **0.144** ($\pm$0.0436) |
| Weather | 0.135 ($\pm$0.0365) | **0.130** ($\pm$0.0348) | **0.130** ($\pm$0.0347) |
| Electricity | **0.120** ($\pm$0.0179) | **0.120** ($\pm$0.0178) | **0.120** ($\pm$0.0179) |
| Avg. Rank | 1.71 | 1.42 | **1.00** |

## 5 Results

We present comparative results of our approach for Chronos in Table 5 which shows the performance of mini T5 Chronos model variant in zero-shot setting along with different fine-tuning settings. We report MASE averaged over 5 runs for both zero-shot and default fine-tuning setting. For AT4TS, we run 10 trials and report MASE corresponding to the best LoRA configuration, which is also averaged over 5 runs. We observe that the performance of our approach is better than zero-shot and default LoRA fine-tuning for 6 out of 10 datasets as highlighted in the last column of Table 5. This can be attributed to the superior performance of LoRa when coupled with automated hyperparameter optimization.

We observe that, for target domain datasets such as Exchange Rate and Australian Electricity which do not share any similarity with the pre-training source datasets, our approach outperforms with a significant margin. To support our argument, we measure dataset similarity between pre-training and target datasets using Maximum Mean Discrepancy (Wang et al., 2021) along with Principal Component Analysis (PCA) (Abdi & Williams, 2010) (See details in Appendix 7.1). However, it is important to note that dataset similarity is not the only criterion that influences the zero-shot and fine-tuned performances of the pre-trained time series foundation models. Based on the findings in (Ekambaram et al., 2024) Section 4.7

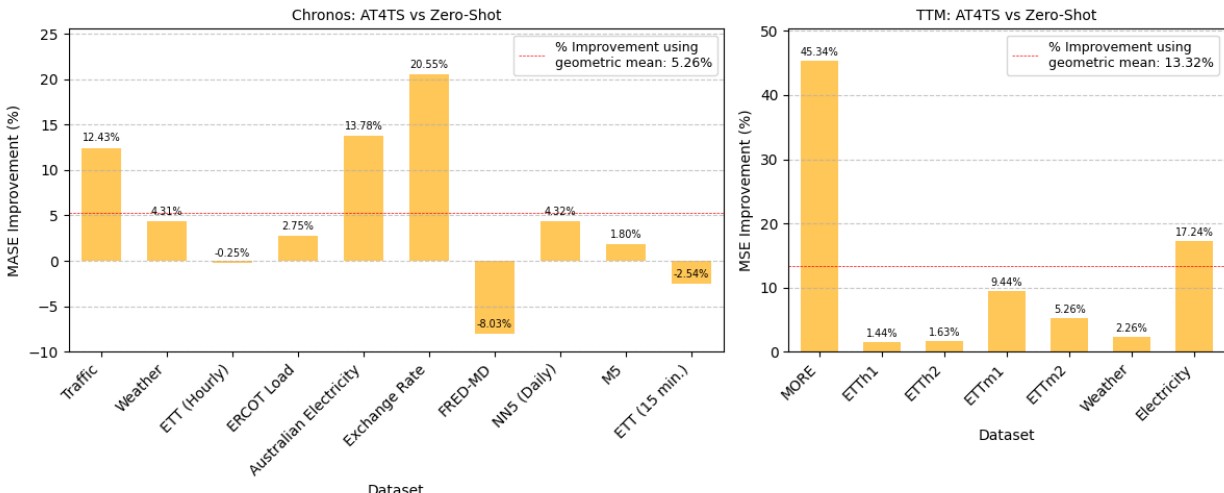

Figure 3: % improvement using geometric mean achieved by AT4TS over zero-shot across datasets for Chronos and TTM.

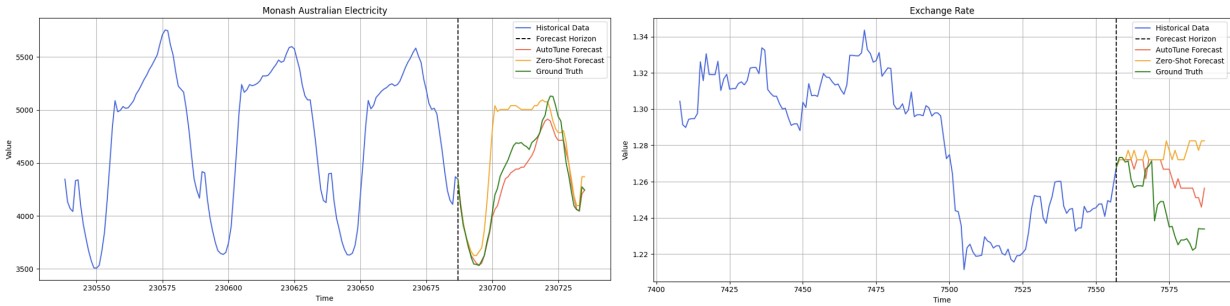

Figure 4: Performance comparison for Zero-shot and Autotuned models for Monash Australian Electricity and Exchange Rate datasets.

(Ablation studies) and (Ansari et al., 2024) section 5.6 (TSMixup Augmentations), dataset coverage and resolution diversity have also shown to play a pivotal role in improving the zero-shot performance of these pre-trained models on unseen (out-of-domain) datasets.

Our results also validate that LoRA achieves superior performance while significantly reducing the number of trained parameters for most of the out-of-domain target datasets. We compare our method against the zero-shot performance using relative error scores. In Figure 3, left plot visualizes the improvement achieved by AT4TS in comparison to the zero-shot model for each dataset. The relative error scores are then aggregated across all datasets using the geometric mean similar to (Ansari et al., 2024). Therefore, our method achieves an overall relative improvement of 5.26% over the zero-shot model performances across all datasets. This is denoted by the red dotted line in the plot. We can see that the autotuned model outperforms the zero-shot model for most datasets. Moreover, it exhibits particularly strong performance on datasets such as exchange rate with a significant MASE improvement of 20.55%. This can be partially attributed to its relative dissimilarity with the pretraining datasets (see the Appendix Section 8.1).

Figure 4 illustrates the predictive performance of the various Chronos models on the target domain datasets : Monash Australian Electricity and Exchange Rate respectively. Here, we compare the forecasts obtained

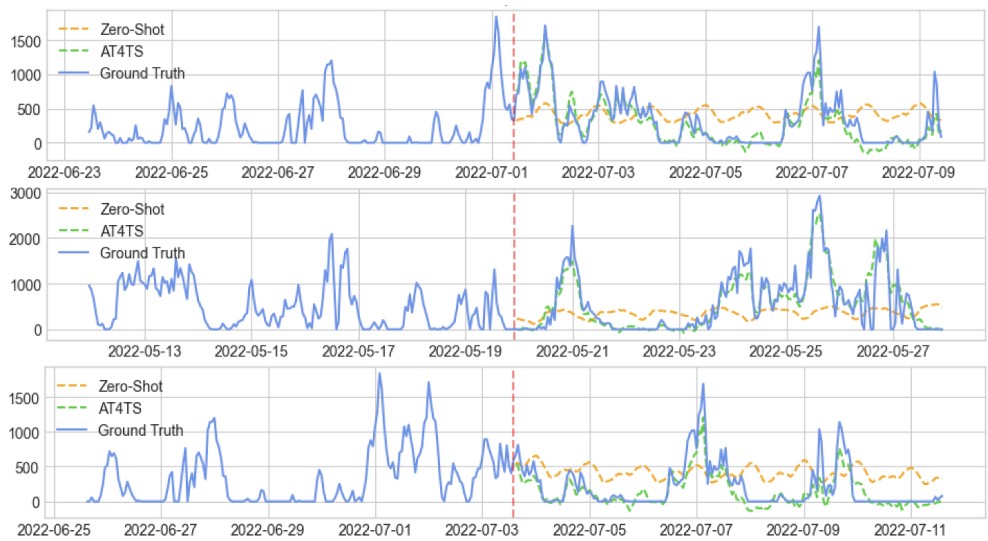

Figure 5: Performance comparison for Zero-shot TTM and autotuned TTM using AT4TS for MORE data.

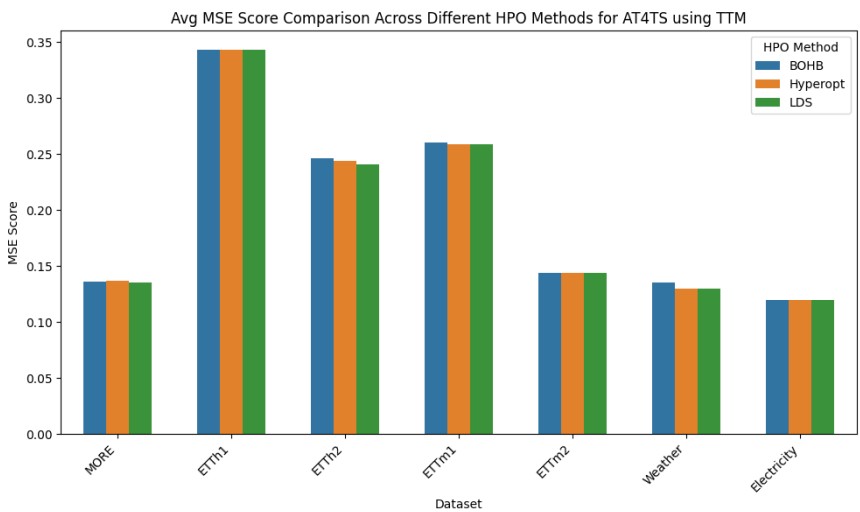

Figure 6: Avg MSE score comparison across different HPO Methods for AT4TS using TTM.

on the test split by our autotuned model with zero-shot models, clearly highlighting the improved prediction accuracy obtained using our approach. In summary, our findings demonstrate that AT4TS can efficiently autotune time series foundation models enhancing their downstream performance in the target domain.

Table 6 further explores the effectiveness of LDS in comparison to other state-of-the-art HPO techniques, namely BOHB and Hyperopt. We observe that the average rank for HyperOpt and LDS is equal. However, in comparison to BOHB, LDS is marginally better with a lower average rank. This demonstrates that LDS performs on par with the state-of-the-art HPO techniques. It is noteworthy that we only run 10 trials in our experiments for Chronos to demonstrate the robustness of AT4TS in a resource constraint environment and achieve strong performance in comparison to zero-shot and default LoRA fine tuning. However, in the presence of more computational and time resources, AT4TS can easily scale well to explore a much larger hyperparameter search space.

Next, we present the results obtained on using TTM-Base model with AT4TS, zero-shot and default fine-tuning approaches for 7 datasets. For default fine-tuning setting, we fix the parameters such as batch size and head dropout based on the target dataset similar to (Ekambaram et al., 2024). The most striking results are observed with the MORE dataset. As shown in Table 7, we observe that the autotuned model using our approach, AT4TS, outperforms both zero-shot and default fine-tuning approaches for all datasets. The predictive performance of autotuned model using AT4TS is better than zero-shot by 45.34% for MORE data as shown in the right plot of Figure 3. The same is illustrated in Figure 5 where we observe significant improvement in the prediction accuracy. Our approach also demonstrates superior performance compared to default fine-tuning, achieving an average MSE of 0.135 with a 5.48% overall improvement across forecast lengths and up to 13% improvement for longer forecast horizons. We highlight that AT4TS successfully explores the best hyperparameter configurations leading to improved fine-tuned performance on out-of-domain target datasets overcoming the drawbacks associated with traditional fine-tuning techniques with fixed hyperparameters.

Similar to Chronos, Table 8 shows the comparison of LDS to other state-of-the-art HPO techniques, namely BOHB and Hyperopt. In general, we observe that HPO methods tend to consistently improve the performance of the fine-tuned model, compared to default fine-tuning configuration. Furthermore, we also calculate the average rank for the 3 HPO methods across all datasets to clearly highlight that LDS is on par with state-of-the-art HPO techniques such as Hyperopt and BOHB, performing slightly better in some cases. The findings are particularly interesting as we can observe from the bar plot in Figure 6 that all HPO methods converge to the lowest MSE corresponding to the best hyperparameter configuration except for datasets, MORE and ETTh2 where LDS is marginally better than other HPO methods. This behaviour can be attributed to the following reasons, *first*, TTM-Base is a much smaller (1M) model in comparison to Chronos-mini (16M) model which implies fewer parameters to fine-tune. *Second*, in the custom fine-tuning approach for TTM, the model backbone is frozen and only the model head is updated. And, *Third*, we run 15 trials tuning a much smaller hyperparameter search space in comparison to Chronos which results in faster convergence to the optimal configuration.

## 6 Conclusion and Future Work

Time series foundation models efficiently capture long-range patterns and dependencies, improving the model's ability to predict complex temporal relationships. However, their successful application to specific downstream tasks needs adaptation to the target domain datasets which can be achieved via fine-tuning. In this work, we propose AT4TS, Autotune for time series foundation models using parameter efficient fine-tuning methods along with LDS as the search strategy. We also compare LDS with other SOTA HPO methods namely BOHB and Hyperopt and show its strong competence. Our approach outperforms traditional fine-tuning strategies specifically for out-of-domain datasets not seen during the model pre-training phase. In addition, we show that AT4TS can be easily applied to both transformer and non transformer-based architectures making it highly scalable yielding competitive results for out-of-domain datasets. In the future, we aim to extend AT4TS for more complex TSFMs and hybrid fine-tuning approaches as we witness continued research in the rapidly evolving TSFM space.

**Acknowledgments**

This work has been partially supported by the 6G-XCEL project (grant agreement 101139194), funded by the EU Horizon Europe program.

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

# 7 Appendix

## 7.1 Dataset Similarity Experiments

In this section, we add the details of the additional experiments performed to measure the similarity between pre-training datasets against the target datasets used in evaluating our approach. We use Maximum Mean Discrepancy(MMD) (Wang et al., 2021) to measure the marginal distribution disparity between the pre-training and evaluation datasets. MMD is a non-parametric distance measure between the source and target domains computed by converting the data into a Reproducing Kernel Hilbert Space (RKHS), using the below equation,

$$\mathrm{MMD}[P(X_{\mathrm{SD}}), P(X_{\mathrm{TD}})] = \|\mathbb{E}[\varphi(X_{\mathrm{SD}})] - \mathbb{E}[\varphi(X_{\mathrm{TD}})]\|_{\mathcal{H}}^2 = \left\| \frac{1}{N_{\mathrm{SD}}} \sum_{p=1}^{N_{\mathrm{SD}}} \varphi(x_p) - \frac{1}{N} \sum_{q=1}^{N} \varphi(x_q) \right\|_{\mathcal{H}}^2$$

where $P(X_{SD})$ and $P(X_{TD})$ are the marginal data distributions of the source and target domains, respectively, $N_{SD}$ and $N$ are the number of data samples in the source and target domains, $\varphi$ is the mapping function from the original feature space to the RKHS, and $\mathcal{H}$ is the RHKS space . The value of MMD starts from 0, indicating that the two domains are completely identical. As the values increase, this indicates the increase in dissimilarity between datasets.

Due to the diverse sizes of pre-training datasets (ranging from 12*1320 to 225280*350640) of these foundation models, we first perform Principal Component Analysis (PCA) (Abdi & Williams, 2010) to reduce the dimensionality and transform the feature space to principal components which is then used to calculate the MMD values. For datasets where the time series length is longer than 10000, we randomly sample 10000 points to perform PCA. For all other datasets, we use the complete dataset for PCA. Table 9 and 10 list the dataset sizes considered for performing PCA for both pre-training and target datasets. The datasets were normalized to have zero mean and unit variance prior to performing PCA. Based on experimentation with different component values, we select 5 principal components for all datasets.

Table 9: Dataset size used for PCA of pre-training datasets.

| Pre-training Datasets | Dataset size for PCA (Time Series length*No. of series) |
|---|---|
| Mexico City Bikes | 10000*494 |
| Solar (5 min) | 10000*5166 |
| Solar (Hourly) | 8760*5166 |
| Taxi (Hourly) | 734*2428 |
| Wind Farms (Daily) | 354*337 |
| Wind Farms (Hourly) | 8514*337 |

Table 10: Dataset size used for PCA of target datasets.

| Target Datasets | Dataset size for PCA (Time Series length*No. of series) |
|---|---|
| Monash Traffic | 10000*862 |
| ETT(Hourly) | 10000*14 |
| ERCOT Load | 10000*8 |
| Australian Electricity | 10000*5 |
| Exchange Rate | 7588*8 |
| FRED-MD | 728*107 |
| NN5 | 791*111 |
| ETT(15 min) | 10000*14 |
| M5 | 1562*30490 |
| Weather | 10000*3010 |

Table 11: MMD values obtained on comparing the Chronos target datasets with a subset of pre-training datasets.

| Target Dataset | Subset of Pre-training Datasets (Chronos) | | | | | | Avg MMD |
|---|---|---|---|---|---|---|---|
| | Mexico City Bikes | Solar (5 min) | Solar (Hourly) | Taxi (Hourly) | Wind Farms (Daily) | Wind Farms (Hourly) | |
| Monash Traffic | 0.066671 | 0.393231 | 0.304424 | 0.060626 | 0.083929 | 0.080981 | 0.1650 |
| Weather | 0.034445 | 0.336337 | 0.257396 | 0.013908 | 0.056538 | 0.047534 | 0.1244 |
| ETT(Hourly) | 0.484774 | 0.824982 | 0.735012 | 0.491731 | 0.437348 | 0.421104 | 0.5658 |
| ERCOT Load | 0.658415 | 1.00162 | 0.91165 | 0.668411 | 0.615951 | 0.583416 | 0.7399 |
| Australian Electricity | 0.716793 | 1.058788 | 0.968819 | 0.725577 | 0.669882 | 0.645624 | 0.7976 |
| Exchange Rate | 0.594259 | 0.936563 | 0.846594 | 0.603339 | 0.547911 | 0.523279 | 0.6753 |
| FRED-MD | 0.096956 | 0.459704 | 0.369734 | 0.12633 | 0.118785 | 0.092587 | 0.2107 |
| NN5 | 0.145958 | 0.515722 | 0.425748 | 0.182453 | 0.163181 | 0.126763 | 0.2600 |
| M5 | 0.038404 | 0.347983 | 0.258042 | 0.014831 | 0.059337 | 0.050844 | 0.1282 |
| ETT(15 min) | 0.450568 | 0.793353 | 0.703383 | 0.460112 | 0.407914 | 0.382236 | 0.5329 |

Table 11 shows the MMD values obtained by comparing the pre-training datasets with the target datasets. We compare each dataset used to validate AT4TS to a subset of pre-training datasets of the Chronos models.

We consider a subset of pre-training datasets from the Chronos paper (Ansari et al., 2024) mentioned in Table 3 Appendix section B. We calculate MMD values per dataset and then provide an average across all the pre-training datasets. We observe that MMD values support our argument for both Australian Electricity and Exchange Rate datasets which show high MMD values of 0.798 and 0.675 respectively indicating relatively high dissimilarity with the pre-training datasets compared with other target domain datasets. Hence, Autotune leads to significantly better results for these datasets given they are both out-of-domain as well as showing dissimilarity with the pre-training datasets.

However, we hypothesize that dataset similarity is not the only criteria that influences the zero-shot and fine-tuned performances of the pre-trained time series foundation models as discussed in (Ekambaram et al., 2024) Section 4.7 (Ablation studies) and (Ansari et al., 2024) section 5.6 (TSMixup Augmentations) where the dataset coverage and resolution diversity have shown to improve the zero-shot performance of these pre-trained models on unseen datasets. These findings point towards other factors beyond distance-based similarity that affect the final performance of these models. As we can see, MMD values only partially support the argument as expected and should not be considered the only deciding factor in assessing the performance improvements achieved on auto-tuning these models.

