# OpenReview forum: "AT4TS : Autotune for Time Series Foundation Models"
_TMLR — Accepted by TMLR_

### Review · Reviewer_y4UW · 2025-07-03

**Summary Of Contributions:**

This paper presents a method for tuning the hyper-parameters of the fine-tuning procedure for time series foundation models on target data. The novelty of the work does not lie in the method itself, but rather in the combination of standard hyper-parameter tuning strategies with standard fine-tuning methods (such as LoRA or forecasting head fine-tuning) applied to standard time series forecasting tasks using standard architectures. The main contribution is therefore the integration and evaluation of these existing components together.

**Audience:**

Yes

**Broader Impact Concerns:**

--

**Claims And Evidence:**

Yes

**Requested Changes:**

First, this paper discusses autotuning strategies for fine-tuning time series foundation models. It would hence be beneficial to provide a review of the state of the art for autotuning fine-tuning methods in fields other than time series forecasting.

Second, as discussed lightly in the experiment section, fine-tuning is likely to behave very differently depending on whether the target domain (or a very similar task) was present in the training set of the model or not.
Whether the target domain was present in the training data is already discussed, but it should be clearly highlighted in the result tables (eg. have separate sections for in-domain and out-of-domain settings).
Then, for domains that are supposed to be "close to" some of the training domains, this claim should be backed by some measure of closeness.

Finally, I have two minor remarks:

* Table 8 should have bold to indicate winners and ex-aequos
* You state that you use two different metrics for the evaluation: why not stick to the same evaluation metric for all your experiments?
    * Also you call MASE the Mean Absolute Squared Error, I guess you meant Mean Absolute Scaled Error

**Strengths And Weaknesses:**

**Strengths:**

* Interesting real-world problem
* The choice of the methods to be benchmarked is sound


**Weaknesses:**

* The paper's novelty is limited and its conclusions are expected (using standard hyper-parameter selection helps even at the fine-tuning stage for time series foundation models)
* Some experimental details could be improved
* Tables 6 and 8 show that there is no clear winner among the hyper-parameter selection methods

---

> ### Author Response · Authors · 2025-07-24
>
> Thank you for your review comments and feedback on our work. Please find below the updated changes in the revised paper.
>
> Requested Changes
>
> RC1 - We have addressed this requested change by updating the related work section with a short review of SOTA methods for Auto-tuning fine-tuning methods in other domains (NLP and image domains).
>
> RC2 - We are in the process of performing additional experiments with a few similarity measures and should be able to provide the analysis as soon as possible.
>
> Minor Comment 1 :
> Table 8 has been updated in the paper to highlight the winners in Bold. In order to clearly showcase the winners across HPO methods for both Chronos and TTM (Table 6 and 8), we have added the average rank across all datasets which clearly demonstrate that LDS is the best (TTM) or at par (Chronos) with other HPO methods.
>
> Minor Comment 2 :
> Thanks for the correction. In our evaluation, MASE refers to Mean Absolute Scaled Error. This has been updated in the paper. We use different evaluation metrics to ensure consistency with the original work that introduced the Time Series Foundation Models used in our paper (MASE for Chronos and MSE for TTM).
>
> If there are any further changes or clarifications needed in the meantime, please let us know.

---

> > ### Author Response · Authors · 2025-07-30
> >
> > We thank the reviewer for raising issue of dataset similarity as additional experiments have provided us with important insights which we share below:
> >
> > RC2-
> > First, we would like to clarify that all the datasets used in the experiments are out-of-domain as none of these datasets were present in the pre-training suite of the models (Chronos and TTM) as discussed in the Section 4 under the Datasets paragraph as follows:
> >
> > “We use these datasets as they have not been used in the pre-training phase of the Chronos T5 models. Similarly, Table 3 lists the datasets used for TTM in our experiments. All the datasets except MORE were used in (Ekambaram et al., 2024) for zero-shot/few-shot evaluation with no overlap with the pre-training datasets.”
> >
> > Hence, we keep the results table unchanged as all the datasets are out-of-domain and we are not using any in-domain dataset that the models have seen during the pre-training phase.
> >
> > Second, based on your feedback on providing a measure of closeness between datasets, we performed additional experiments using Maximum Mean Discrepancy (MMD) (Wang et al., 2021) measure as the criteria to measure the closeness of source(pre-training) and target domain datasets. MMD is used to measure the marginal distribution disparity between the pre-training and evaluation datasets. It is a non-parametric distance measure between the source and target domains computed by converting the data into a Reproducing Kernel Hilbert Space (RKHS), using the below equation:
> >
> > $$\text{MMD}[P(X_{\text{SD}}), P(X_{\text{TD}})] = \left\| \frac{1}{N_{\text{SD}}} \sum_{p=1}^{N_{\text{SD}}} \varphi(x_p) - \frac{1}{N_{\text{TD}}} \sum_{q=1}^{N_{\text{TD}}} \varphi(x_q) \right\|_{\mathcal{H}}^2$$
> >
> > where $P(X_{SD})$ and $P(X_{TD})$ are the marginal data distributions of the source and target domains, respectively, $N_{SD}$ and $N_{TD}$ are the number of data samples in the source and target domains, $\varphi$ is the mapping function from the original feature space to the RKHS, and $\mathcal{H}$ is the RHKS space.
> >
> > The value of MMD starts from 0, indicating that the two domains are completely identical. As the values increase, it signifies the increasing dissimilarity between datasets.
> >
> > Due to the diverse sizes of pre-training datasets (ranging from 12×1320 to 225280×350640 ) of these foundation models, we first perform Principal Component Analysis (PCA) to reduce the dimensionality and transform the feature space to principal components which is then used to calculate the MMD values. For datasets where the time series length is longer than 10000, we randomly sample 10000 points to perform PCA. For all other datasets, we use the complete dataset for PCA. Based on experimentation with different component values, we select 5 principal components for all datasets. Please find the detailed experiment results in the table below:
> >
> > |**Pre-training Datasets→  Target Datasets↓** | **Mexico City Bikes** | **Solar (5 min)** | **Solar (Hourly)** | **Taxi (Hourly)** | **Wind Farms (Daily)** | **Wind Farms (Hourly)** | **Avg MMD** |
> > |---|---|---|---|---|---|---|---|
> > | Monash Traffic |0.067 | 0.393 | 0.304 | 0.061 | 0.084 | 0.081 | 0.165 |
> > | Weather | 0.034 | 0.336 | 0.257 | 0.014 | 0.057 | 0.048 | 0.124 |
> > | ETT(Hourly)| 0.485 | 0.825 | 0.735 | 0.492 | 0.437 | 0.421 | 0.566 |
> > | ERCOT Load| 0.658 | 1.002 | 0.912 | 0.668 | 0.616 | 0.583 | 0.740 |
> > | Australian Electricity | 0.717 | 1.059 | 0.969 | 0.726 | 0.670 | 0.646 | 0.798 |
> > | Exchange Rate | 0.594 | 0.937 | 0.847 | 0.603 | 0.548 | 0.523 | 0.675 |
> > | FRED-MD | 0.097 | 0.460 | 0.370 | 0.126 | 0.119 | 0.093 | 0.211 |
> > | NN5 | 0.146 | 0.516 | 0.426 | 0.182 | 0.163 | 0.127 | 0.26 |
> > | M5 | 0.038 | 0.348 | 0.258 | 0.015 | 0.059 | 0.051 | 0.128 |
> > | ETT(15 min) | 0.451 | 0.793 | 0.703 | 0.460 | 0.408 | 0.382 | 0.533 |
> >
> > Please refer to the remaining discussion in the following comments (max characters reached).

---

> > > ### Author Response · Authors · 2025-07-30
> > >
> > > *Discussion:*
> > >
> > > We consider a subset of pre-training datasets from the Chronos paper (Ansari et al., 2024) mentioned in Table 3 Appendix section B. We calculate MMD values per dataset and then provide an average across all the pre-training datasets. We observe that MMD values support our argument for both Australian Electricity and Exchange Rate datasets which show high MMD values of 0.798 and 0.675 respectively indicating relatively high dissimilarity with the pre-training datasets compared with other target domain datasets. Hence, Autotune leads to significantly better results for these datasets given they are both out-of-domain as well as showing dissimilarity with the pre-training datasets.
> > >
> > > However, we hypothesize that dataset similarity is not the only criteria that influences the zero-shot and fine-tuned performances of the pre-trained time series foundation models as discussed in (Ekambaram et al., 2024) Section 4.7 (Ablation studies) and (Ansari et al., 2024) section 5.6 (TSMixup Augmentations) where the dataset coverage and resolution diversity have shown to improve the zero-shot performance of these pre-trained models on unseen datasets. These findings point towards other factors beyond distance-based similarity that affect the final performance of these models. We would like to mention that we have been cautious in our experimental analysis while attributing the performance gains achieved using autotune solely to dataset similarity. As we can see, MMD values only partially support the argument as expected and should not be considered the only deciding factor in assessing the performance improvements achieved on auto-tuning these models.
> > >
> > > We only do this analysis with Chronos pre-training and target datasets for now and shall extend it to TTM if needed.
> > >
> > > We sincerely thank you for your insightful comments and hope the above experimental analysis has helped in clarifying your concerns. Please let us know if you have any further questions.
> > >
> > > *References:*
> > >
> > > Abdul Fatir Ansari, Lorenzo Stella, Caner Turkmen, Xiyuan Zhang, Pedro Mercado, Huibin Shen, Olek-
> > > sandr Shchur, Syama Sundar Rangapuram, Sebastian Pineda Arango, Shubham Kapoor, et al. Chronos:
> > > Learning the language of time series. arXiv preprint arXiv:2403.07815, 2024.
> > >
> > > Vijay Ekambaram, Arindam Jati, Nam H Nguyen, Pankaj Dayama, Chandra Reddy, Wesley M Gifford,
> > > and Jayant Kalagnanam. Ttms: Fast multi-level tiny time mixers for improved zero-shot and few-shot
> > > forecasting of multivariate time series. arXiv preprint arXiv:2401.03955, 2024.
> > >
> > > Wei Wang, Baopu Li, Shuhui Yang, Jing Sun, Zhengming Ding, Junyang Chen, Xiao Dong, Zhihui Wang,
> > > and Haojie Li. A unified joint maximum mean discrepancy for domain adaptation. arXiv preprint
> > > arXiv:2101.09979, 2021.

---

> > > > ### Comment · Reviewer_y4UW · 2025-08-05
> > > > **Post-rebuttal**
> > > >
> > > > I would like to thank the authors for taking my requested changes into consideration. It is a bit unclear to me wether they plan to include the similarity between datasets in the paper, in the supplementary material, or nowhere.

---

> > > > > ### Author Response · Authors · 2025-08-05
> > > > >
> > > > > Thank you for the question. We would like to clarify that we have now added the details of the extended experiments and the related learnings in the Appendix section of the paper where we added the discussion on distance measures as well as dataset similarity discussions from literature and the corresponding correlations with autotuned performance of pre-trained TSFMs. The core of this appendix section comes from the comments as described above. We have also updated the Results section in the main paper to highlight these new findings based on dataset similarity experiments.
> > > > > Please let us know if you have any further questions.

---

### Review · Reviewer_CEoN · 2025-07-08

**Summary Of Contributions:**

- The authors proposed an end-to-end fine-tuning method for time-series forecasting framework with hyper-parameter optimization.
- The authors provided various experiments with diverse combinations of time-series foundation models as well as hyper-parameter optimization methods.
- The results show the following conclusion in terms of performance: fine-tuning with hyper-parameters optimization > fine-tuning > zero-shot.

**Audience:**

Yes

**Claims And Evidence:**

Yes

**Requested Changes:**

See the above weakness part.

**Strengths And Weaknesses:**

**Weakness**
- In general, it is a little bit difficult to understand the contributions of this paper. This paper propose an end-to-end time-series forecasting model fine-tuning with hyper-parameter optimization which is more like combining the fine-tuning techniques with hyper-parameter optimization on time-series forecasting domains.
- Also, the results are quite trivial. Zero-shot < fine-tuning with default hyper-parameters < fine-tuning with hyper-parameter optimization. This is quite trivial on not only the time-series forecasting, but also any predictive model cases.
- In general, to make the customized time-series model, researchers and developers usually follow the similar procedure: (i) fine-tuning with LoRA or other PEFT, (ii) hyper-parameter optimization. It is unclear how the proposed framework is different (and novel) from the standard procedure.
- Minor: Citation format (and style) in the entire paper needs to be modified (please use parenthetical citation)

---

> ### Author Response · Authors · 2025-07-24
>
> Thank you for your review comments and feedback on our work. Please find below the clarifications for several important points raised :
>
> W1 - We sincerely acknowledge your concerns and would like to highlight the important contributions of our work in the context of Time Series Foundation Models. To the best of our knowledge, autotuning has not been applied in the context of fine-tuning Time Series Foundation Models yet.  Since our work is the first attempt of autotuning, we highlight the value it adds to the predictive performance of these models compared to zero-shot/default fine tuning approaches.
>
> W2 - Our results highlight that automated fine-tuning using HPO techniques can yield much significant performance improvements compared to both zero-shot and default fine-tuning settings in most cases. However, we also encounter scenarios where autotuning might not be of significant value especially when the target datasets are similar to the pre-training datasets. This demonstrates an important finding that auto-tuning does not always lead to significant improvement over zero-shot.
>
> W3 - We agree that fine-tuning a time series model followed by HPO is a general practice, however, it might not always be applicable in all scenarios. AT4TS provides an end-to-end solution for auto-tuning TSFMs benchmarking different HPO techniques including the custom LDS approach. We believe our work is a novel contribution in the context of TSFMs as there is no prior research that solves this real-world problem. Moreover, we would like to reiterate that we further add value to AT4TS by providing both, a model agnostic as well as fine-tuning method agnostic approach for general applicability.
>
> Minor - We have updated the citation style used in the paper to parenthetical citation.
>
> If there are any further changes or clarifications needed, please let us know.

---

### Review · Reviewer_uxTe · 2025-07-18

**Summary Of Contributions:**

Time series foundation models (TSFMs) are able to produce accurate forecasts in zero-shot manner. However, it has been repeatedly been shown that fine-tuning these models on the target task often leads to even better performance. The specific configuration for fine-tuning, such as choice of the learning rate or batch size, can significantly affect the final performance. This paper address this problem, and introduces an automated hyperparameter selection procedure for fine-tuning of TSFMs. The proposed approach is based on the Limited Discrepancy Search methods. Empirical evaluation demonstrates the effectiveness of this approach, as it often leads to better forecast accuracy compared to zero-shot forecasting or fine-tuning with the default configuration. The results are validated using two TSFM architectures (Chronos-mini and TTM), various datasets and evaluation metrics.

**Audience:**

Yes

**Broader Impact Concerns:**

No concerns.

**Claims And Evidence:**

Yes

**Requested Changes:**

- The ablation study comparing LDS with other existing HPO strategies (Tables 6 and 8) does not convincingly show that the LDS strategy consistently outperforms the other HPO methods. The errors bars for the methods overlap, and no method emerges as universally superior to other. A more appropriate conclusion based on these results could be that HPO methods tend to consistently improve the performance of the fine-tuned model, compared to fine-tuning with the default configuration.
- Using arithmetic mean when aggregating the relative improvements over the zero-shot model in Section 5 / Figure 3 is a questionable choice since we are effectively dealing with rations, and the relative improvement (1 - error_method / error_baseline) can take values in (-\infty, 1). A more appropriate choice would be the geometric mean of the relative errors error_method / error_baseline.

**Strengths And Weaknesses:**

**Strengths**
- The paper studies an interesting and relevant question - should we fine-tune TSFMs, and if so, how it should be done? The findings that 1/ fine-tuning consistently improves the forecast accuracy and 2/ the specific configuration used during fine-tuning can noticeably affect the results are relevant for the research community.
- The experiments are designed in a way that clearly answers the research questions posed in the paper. The effectiveness of the proposed approach is demonstrated across different model architectures (encoder-decoder transformer / tsmixer), fine-tuning strategies (LoRA / output layer fine-tuning), evaluation metrics (MASE / WQL) and dataset types.

**Weaknesses**
I have not identified major weaknesses related to the submission, but there are a few claims and methodological choices that could be adjusted to further strengthen the work. Please see "Requested Changes" below.

**Minor comments**
- Consider using the `\citep` command for citations that should be put in parenthesis.
- A potentially relevant work that also investigates fine-tuning strategies for TSFMs including Chronos: Arango et al., "ChronosX: Adapting Pretrained Time Series Models with Exogenous Variables", AISTATS 2025

---

> ### Author Response · Authors · 2025-07-24
>
> Thank you for your review comments and feedback on our work. Please find below the updated changes in the revised paper.
>
> Minor comments
>
> - We have updated the citation style as requested.
>
> - We would like to thank you for sharing the paper titled “ChronosX : Adapting Pre-Trained Time Series Models with Exogenous Variables”, it highlights a different approach to fine-tuning Chronos models using covariates. However, the paper focusses on the inclusion of external information by means of covariates into the forecasting models such as Chronos and evaluates the performance improvement. Our work uses TTM model which achieves multivariate forecasting by incorporating exogenous variables as part of their architecture design. We believe using ChronosX in our framework can provide a promising future work direction to explore the added benefits of applying AT4TS for automated fine-tuning of the covariate adapter modules.
>
> Requested changes:
>
> For RC1 , we have addressed the requested change by updating Table 6 and 8 with Average rank of HPO methods across datasets which shows that LDS is at par (Chronos) or better (TTM) than other HPO methods. Furthermore, we also agree to your proposed conclusion that HPO methods consistently improve the fine-tuned performance compared to default fine-tuning configurations and updated the discussion in the results section.
>
> For RC2, as suggested, we have updated Figure 3 in the revised paper with the geometric mean of the relative error scores and added the justification of using geometric mean in the text discussing the figure. Geometric mean proves to be a more appropriate way to aggregate such relative scores.
>
> If there are any further changes or clarifications needed, please let us know.
> Thank you once again for your comments and for your time!

---

### Decision · Action_Editor_6i51 · 2025-08-15

**Recommendation:** Accept with minor revision

**Additional Comments:**

This manuscript studies autotuning for time series foundation models. It follows the common practice of tuning foundation models, and shows performance improvement by autotuning. The revised version has clarified and addressed most of the concerns raised by the reviewers. Two reviewers still hold the opinion that the manuscript does not contribute novel technologies to the field. Nontrivial insights are absent which are expected to dive into the special behaviors of time series foundation models when autotuning. Nonetheless, the paper is self-contained in the context it is targeted, and the presentation is clear with sufficient details that will facilitate future research in this direction.

With the minor revision, the paper is expected to provide more insights/discussions into the behaviors of time series foundation models when performing fine-tuning. The difference from standard SFT on LLMs shall be highlighted in the new version. Also, please address the minor concerns raised by the reviewers regarding the improper claiming of the empirical findings.

**Audience:**

Yes

**Audience Explanation:**

The time series analysis community has witnessed a mainstream of time series foundation models (TSFMs). This work established a study on autotuning the TSFMs and showed empirical benefits, which will be interested by the community.

**Claims And Evidence:**

Yes

**Claims Explanation:**

As mentioned by the reviewers, some of the claims in the empirical study have not been well supported. Section 5 remains inaccurate:

Table 6 further demonstrates the effectiveness of LDS in comparison to other state-of-the-art HPO techniques, namely BOHB and Hyperopt. LDS clearly outperforms by a significant margin in 4 out of 10 datasets

The average rank of LDS exactly equals the average rank of HyperOpt, and HyperOpt similarly outperforms other methods by a significant margin on 4 out of 10 datasets.